# Making the BEST decision-the BESTa project development, implementation and evaluation of a digital Decision Aid in Swedish cancer screening programmes- a description of a research project

**Kaisa Fritzell[1,2], Berith Hedberg[3], Anke Woudstra[4], Anna Forsberg[5], Marika Sventelius[6], Anders Kottorp[7], Anna Jervaeus[1]** *

1 Department of Neurobiology, Care Sciences and Society, Division of Nursing, Karolinska Institutet, Huddinge, Sweden, 2 Hereditary Cancer Clinic, Theme Cancer, Karolinska University Hospital, Stockholm, Sweden, 3 School of Health Sciences, Jönköping University, Jönköping, Sweden, 4 Team Advies en Onderzoek, Municipal Health Service (GGD) Kennemerland, Haarlem, the Netherlands, 5 Department of Medicine, Solna, Karolinska Institutet, Stockholm, Sweden, 6 Regional Cancer Centre, Stockholm, Gotland, Sweden, 7 Faculty of Health and Society, Malmö University, Malmö, Sweden

* anna.jervaeus@ki.se

## Abstract

### Background

Sweden has a long tradition of organized national population-based screening programmes. Participation rates differ between programmes and regions, being relatively high in some groups, but lower in others. To ensure an equity perspective on screening, it is desirable that individuals make an informed decision based on knowledge rather than ignorance, misconceptions, or fear. Decision Aids (DAs) are set to deliver information about different healthcare options and help individuals to visualize the values associated with each available option. DAs are not intended to guide individuals to choose one option over another. The advantage of an individual Decision Aid (iDA) is that individuals gain knowledge about cancer and screening by accessing one webpage with the possibility to communicate with health professionals and thereafter make their decision regarding participation. The objective is therefore to develop, implement and evaluate a digital iDA for individuals invited to cancer screening in Sweden.

### Methods

This study encompasses a process-, implementation-, and outcome evaluation. Multiple methods will be applied including focus group discussions, individual interviews and the usage of the think aloud technique and self-reported questionnaire data. The project is based on The International Patient Decision Aid Standards (IPDAS) framework and the proposed model development process for DAs. Individuals aged 23–74, including women (the cervical-, breast- and CRC screening module) and men (the CRC screening module), will be

relevant data from this study will be made available upon study completion.

**Funding:** The project is funded by the Regional Cancer Centre, Stockholm-Gotland, Sweden (KF) VKN 2020-0212 https://cancercentrum.se/stockholm-gotland/ and the Swedish Cancer Society (AJ) 22 0598 FE 01 H. https://www.cancerfonden.se The funders did not and will not have a role in study design, data collection and analysis, decision to publish, or preparation of the manuscript.

**Competing interests:** The authors have declared that no competing interests exist.

**Abbreviations:** DA, decision aid; iDA, : individual decision aid; IPDAS, International Patient Decision Aid Standards; SKR, The Swedish Association of Local Authorities and Regions; RCC, Regional Cancer Centres.

included in the developmental process. Efforts will be made to recruit participants with self-reported physical and mental limitations, individuals without a permanent residence and ethnic minorities.

## Discussion

To the best of our knowledge, the present study is the first attempt aimed at developing an iDA for use in the Swedish context. The iDA is intended to facilitate shared decision making about participation in screening. Furthermore, the iDA is expected to increase knowledge and raise awareness about cancer and cancer screening.

## Patient or public contribution

Lay people are involved throughout the whole development and implementation process of the digital DA.

## Trial registration

NCT05512260.

## Background

Sweden has a long tradition of organized national population-based screening programmes. The first started in the 1960s with a pap smear test to prevent cervical cancer, followed by mammography screening for breast cancer in the 1980s and a regional screening programme (Stockholm-Gotland) with faecal testing for colorectal cancer (CRC) in 2009. The cervical cancer screening programme includes women aged 23–70 and the National Board of Health and Welfare [1] states that that all regions should provide a test for human papillomavirus (HPV) every fifth year in women from 23 years of age, and every seventh year in women from 49 years of age. In recent years, self-sampling test kits for HPV have been offered, especially due to the COVID-19 pandemic. The National Board of Health and Welfare further states that mammography should be provided to women aged 40–74, every 18th to 24th month [1] and they recommend CRC screening with faecal testing biannually for individuals aged 60–74 years [2]. CRC screening has previously been offered to individuals in the Stockholm—Gotland region but are now being introduced in all regions in Sweden.

Screening invitations are sent to targeted individuals by post and the national screening programmes in Sweden should be free of charge. Sweden is divided in 21 regions and each region has responsibility for the health care sector, including organising the cancer screening. The region where the individual lives, sends the invitation. The decision on whether or not to participate is made by the individual. For a cancer screening programme to be effective, a high participation rate is necessary. However, it is well known that screening programmes face challenges in obtaining a high participation rate, thereby jeopardizing the benefits of screening. In Sweden, latest figures from 2021, show an average participation rate of 71% for cervical cancer screening, 81% for breast cancer screening and 71% for CRC screening (Stockholm-Gotland) but with clear socioeconomic differences regarding participation [3]. For CRC screening, s recent follow-up study from Sweden showed that among those with a positive stool test, 91% participate in the subsequent colonoscopy [4]. A low participation rate also jeopardizes the equity of screening programmes, as it is well known that individuals with a lower

socioeconomic status, ethnic minorities and individuals with various limitations participate to a lesser extent [5–9]. Racial and ethnic inequities have also been proven in a recent study from 2023 regarding oncology clinical trial participation [10]. A high participation rate is therefore crucial, not only for preventing cervical-, breast- and CRC in the population, but also for reducing inequalities related to health prevention efforts, in this case population-based screening. However, when approaching seemingly healthy individuals, it is vital to ensure their autonomy [11]. In addition to socioeconomic factors, other reported barriers, irrespective of cancer screening programme, are poor understanding and knowledge of cancer and screening [12–16]. Differences in barriers and facilitators for uptake between the programmes also relate to the screening method. For cervical cancer screening the self-sampling HPV test has been found to be highly acceptable [17] and to increase uptake [18]. In addition, the uptake increased when the faecal sampling test was changed to a more user-friendly sampling tube in CRC screening programmes [19]. In contrast, research shows that information interventions about the relatively high risk of over detection associated with mammography have a negative impact on the willingness to participate in breast cancer screening [20].

Hence, an equity perspective is crucial for the present project and in line with three of the Sustainable Development Goals (SDGs): Good health and well-being, Gender equality and Reduced inequalities. The SDGs are included in Agenda 2030, adopted by UN in 2015 and communicated by our government and authorities [21, 22]. To strengthen the equity perspective on screening it is desirable that individuals make shared decisions based on knowledge rather than ignorance, misconceptions, or fear. Therefore, the present project is based on the theoretical framework of shared decision making (SDM), with the overarching goal of improving the quality of health care decisions. SDM in this project is based on the work of Charles and Gafni [23] with three cornerstones: *information/knowledge*–about cancer, screening, screening tests and pros and cons; *values/preferences*–individuals' attitude and behaviour when invited to screening and *involvemen*t–in the decision of both the individual and health-care professionals.

Our previous research includes both qualitative and quantitative studies exploring how individuals' reason about their decision whether or not to participate in CRC screening. The research was part of the Screening of Swedish Colons (SCREESCO) study (ID: NCT02078804), where individuals were randomized to colonoscopy or faecal immunochemical tests (FIT). The overall aim of SCREESCO was to investigate the effects of CRC screening on the incidence of and mortality due to CRC in Sweden [4]. Our studies found few differences between participants and non-participants. Both groups lacked knowledge about CRC and screening but had adequate health literacy (HL) and low levels of anxiety in relation to their decision [16, 24, 25]. However, the groups differed when measuring values and preferences. Non-participants had a more fatalistic approach, while participants viewed CRC screening as a way of "having control over one's health" [16]. Regarding different screening methods, we analysed individuals' experiences of the CRC screening procedure (colonoscopy and faecal test). The results included both positive and negative emotional reactions, a varying burden of the practical part of the screening procedure, experiences of being inconsistently informed and involved and expectations not matching the reality [26]. To further understand the need for support from healthcare professionals we analysed 2,100 of 10, 000 documented telephone calls (14%) to the SCREESCO helpline (the study provided such support for two hours every day). Unsubscribing or subscribing to screening was the most frequent reason for calling, followed by organisational issues counselling, and faecal test problems. Counselling mainly concerned abnormal faecal test results, including anxiety related to the result, followed by a need to discuss whether or not to participate in the SCREESCO study [27]. Based on our previous findings and the importance of applying an equity perspective on screening, it is desirable that

individuals make an informed decision regarding participation. A suitable format to consider in the present context is a Decision Aid (DA). DAs are set to deliver information about different healthcare options and to help individuals visualize their values associated with available options [28], here screening participation. The goal of the present project is therefore to support individuals in their decision-making process by the development of a digital individual decision aid (iDA). Implementation is planned in all national population-based cancer screening programmes in Sweden, starting with CRC and followed by breast and cervical cancer.

The International Patient Decision Aid Standards (IPDAS) [29] consists of quality criteria which are important to consider when developing DAs [30, 31]. The planned digital iDA will be accessible online with various content and directed towards individuals invited to cancer screening, in order to support shared decision-making. The advantage of an iDA is that individuals gain knowledge about cancer and screening by accessing one webpage with a possibility to communicate with health professionals for advice and support, instead of searching the internet by themselves. Existing evidence shows that people using DAs in connection with a treatment or screening decision increase their knowledge and feel more involved and certain of what matters most to them, i.e. their values [28, 32]. DAs can also be cost-effective, as shown by a decision-analytic model [33]. Nevertheless, DAs are not intended to guide individuals to choose one option over another.

To date, no iDA for any of the screening programmes exists in Sweden. At the same time, the individual is expected to make the decision on whether or not to participate in screening alone which is why this project is warranted. By designing the communication strategies in various ways regarding information about screening options and helping individuals construct, clarify and communicate personal values, our intention is to support the decision-making process and address and acknowledge the equity perspective, by making the effort to approach individuals considered vulnerable and belonging to minority groups in the society. Therefore, the aim is to develop, implement and evaluate a digital individual Decision Aid for people invited to cancer screening in Sweden.

## Methods

### Design

This study encompasses a process, implementation and outcome evaluation. Multiple methods will be used including self-reported questionnaire data, focus group discussions and individual interviews. The project is divided into two phases: Phase 1 consists of the process and initial implementation and Phase 2: the outcome evaluation. The two phases are presented below, please also see the Schedule of enrollment (Fig 1). The work and outline of this paper is inspired by similar research on development of digital decision aids in both Sweden and the Netherlands [34–36].

### The individual Decision Aid (iDA)

A digital iDA with various content (accessible online from a laptop, smartphone or padlet) will be developed for individuals invited to cancer screening. It will also be public and accessible to all who are interested in cancer and screening in Sweden. Digital solutions from authorities and companies are well-developed and frequently used in Sweden. DAs related to screening usually include information on the particular disease, screening, screening tests, benefits and harms and some value clarification exercises (e.g. interactive questions) [28] aimed at shedding light on the individual´s values and preferences, knowledge and lifestyle.

The iDA in the present project will address all three screening programmes with one module for generic content (e.g. cancer, cancer screening) and three separate modules for

Figure. Example template of recommended content for the schedule of enrolment, interventions, and assessments.*

| | STUDY PERIOD | | | | | | | |
|---|---|---|---|---|---|---|---|---|
| | **Enrolment** | **Allocation** | **Post-allocation** | | | | | **Close-out** |
| **TIMEPOINT**** | $-t_1$ | **0** | $t_1$ | $t_2$ | $t_3$ | $t_4$ | *etc.* | $t_x$ |
| **ENROLMENT:** | NA | NA | NA | NA | NA | NA | NA | NA |
| **Eligibility screen** | X | NA | NA | NA | NA | NA | NA | NA |
| **Informed consent** | X | NA | NA | NA | NA | NA | NA | NA |
| *[List other procedures]* | NA | NA | NA | NA | NA | NA | NA | NA |
| **Allocation** | NA | NA | NA | NA | NA | NA | NA | NA |
| **INTERVENTIONS:** | NA | NA | NA | NA | NA | NA | NA | NA |
| *[Intervention A]* | NA | NA | NA | NA | NA | NA | NA | NA |
| *[Intervention B]* | NA | NA | NA | NA | NA | NA | NA | NA |
| *[List other study groups]* | NA | NA | NA | NA | NA | NA | NA | NA |
| **ASSESSMENTS:** | NA | NA | NA | NA | NA | NA | NA | NA |
| *[List baseline variables]* | NA | NA | NA | NA | NA | NA | NA | NA |
| *[List outcome variables]* | nA | NA | NA | NA | NA | NA | NA | NA |
| *[List other data variables]* | NA | NA | NA | NA | NA | NA | NA | NA |

*Recommended content can be displayed using various schematic formats. See SPIRIT 2013 Explanation and Elaboration for examples from protocols.
**List specific timepoints in this row.

**Fig 1. SPIRIT schedule of enrollment.**

screening specific contents that differ with regard to cervical-, breast- and CRC screening. We will start by developing the CRC screening module and then continue with the other two. The iDA is considered to be an informative and interactive tool with the possibility to search for information according to individual preferences.

**Phase 1 Description of the process and implementation evaluation.** *Participants and study setting.* Individuals eligible for screening, i.e. women aged 23–74 (for the cervical, breast

and CRC screening module) and men aged 60–74 (the CRC screening module), will be included. Efforts will be made to recruit individuals with self-reported physical and mental limitations, individuals without a permanent residence and ethnic minorities (with the ability to read and speak the Swedish language) in Sweden. Eligible participants will be approached via advertisements and by contacting relevant Non-Governmental Organisations (NGOs). Lay people will be included in different phases of the project. In addition, various clinical and other experts will be included in the process, when relevant.

*Procedure and outcomes*. The project is based on the IPDAS framework [29, 31] and the proposed model development process for DAs presented by Coulter et al. [30]. For an overview of the procedure see S1 Fig. An overview of the process phase 1.

1. Define scope-a description of breast, cervical and CRC including screening will be formulated, information will be gathered regarding treatment, screening tests, true positive and true negative results, risks and benefits of screening vs. non-screening and detection probabilities. The target audience is all individuals invited to cancer screening.

2. Steering group-a multidisciplinary group will be formed with stakeholders: lay people (aged 23–74), with differing socioeconomic status and ethnicity, from urban and rural areas, invited to and not invited to screening; clinical experts (oncologist, endoscopist, gastroenterologist, gynaecologist, registered nurse, psychologist) and other experts (in psychometry, IT, decision-making and digital tools); collaboration partners (The Swedish Association of Local Authorities and Regions-SKR and The Regional Cancer Centres-RCC and The National Board of Health and Welfare).

3. Design-provision of the iDA in paper format, i.e. a version with the tentative content, including the research and evidence on which the iDA is based, as well as a description of the prototype development. The design of the paper version is inspired by the work of Schwartz and colleagues [37] and its content and comprehensibility in terms of format (information and description in text and images) and language use will be evaluated by lay persons and experts.

4. Alpha testing-the digital iDA (accessible online from a laptop, smartphone or padlet) will be evaluated for its content, comprehensibility, usability and feasibility in terms of format (information and descriptions in text, spoken animations, images, audios, animated videos, certain clickable words to provide additional information and interactive questions to help individuals to clarify and express values), language use; setting (digital) and timing (open to everyone, link will be provided with the invitation).

5. Beta testing-the iDA (accessible online from a laptop, phone or padlet) will be tested in a "real world setting" and evaluated in terms of format (information and descriptions in text, spoken animations, images, audios, animated videos, certain clickable words to provide additional information and interactive questions to help individuals to clarify and express values), language use; setting (digital) and timing (open to everyone, link will be provided with the invitation).

*Data collection*. During steps 3–5 (design, alpha and beta testing) data will be collected using different qualitative methods. Data encompassing content, format, language use, setting and timing of the iDA will be generated from focus group discussion and individual interviews, including the think-aloud technique. An interview guide with open-ended questions and optional probing questions will guide the data collection. Participants will be able to choose to participate in a focus group discussion or an individual interview. The interview will be performed in Swedish and take place online (Zoom or Teams) or in real life. One researcher

performing the interview and one researcher documenting it in a pre-organised Excel sheet. Certain aids, such as an interpreter for deaf individuals, will be provided. Data collection, in each of the three steps, will be ongoing until data saturation has been reached [38], i.e., the availability of enough and rich data displaying both patterns and variations regarding the topic being studied. As one benchmark, the findings by Faulkner [39] on increased samples in usability testing will be applied. The percentage of usability problems increased by number of users, i.e. 5 users found a minimum of 55% problems, 10 users 82% and 20 users 95%. This model was applied in a previous study describing the development of a decision aid for patients with acute Achilles tendon rupture [40]. Another benchmark is a previous study on the same topic where 25 interviews were needed in the alpha phase, until saturation was reached [35]. All individuals included in phase 1 will sign (in written) a consent form prior to inclusion.

During step 3, (design) evaluation of the questions, including the interactive ones (values and preferences, knowledge and lifestyle exercises) will be generated from concurrent cognitive interviewing where the individual gives a verbal account of their thinking while responding to the questions included in the iDA [41]. The interviews will follow a think aloud approach where individuals are asked to respond to the interactive questions and encouraged to think out loud and verbalize their thoughts. This provides an understanding of the perception of each question, as the participants work through the content. In addition, probing questions will be asked when changes in appearance, such as frowning or hesitation, occur and when, or if, the session leader needs further clarification [42].

*Data management*. Regarding demographic data, names will be removed and all participants will be given a unique code. Data will be stored in a secure facility at the Division of Nursing, Karolinska Institutet in accordance with university regulations.

*Data analyses*. Data generated from focus group discussions and individual interviews, will be analysed by applying aspects from both summative content analysis and conventional content analysis, as described by Hsieh and Shannon [43]. The summative approach aims at identifying and quantifying words or context in the data to explore the usage. We have previously applied a similar approach [27]. Conventional content analysis is appropriate when the aim is to describe something, here participants views on the iDA in its different stages. Codes are inductively derived from the data, then sorted and organised, and finally categories are formulated answering the aim [43].

*Participant timeline*. The five steps presented under Procedure and outcomes will be performed during the period 2021–2026.

2021: 1/definition of scope and purpose of the DA and target audience; 2/the formation of a steering group including clinical experts and lay people.

2022–2023: 3/design of the DA while taking account of the views and opinions of lay people and clinical experts regarding the format, setting, timing, clinical evidence and prototype development. This work is ongoing at the moment with data collection in step 3 (design).

2024–2026: 4/alpha testing including comprehensibility and usability; 5/beta testing including feasibility.

*Recruitment*. Clinical and other experts will be approached via the Regional Cancer Centres (RCC), different hospitals and universities in Sweden. They will define the scope, form a steering group and contribute to the design phase, as well as the alpha- and beta testing.

The process for recruiting lay persons, aged 23–74 will be approached in several ways. To reach any ethnic minority, organizations targeting such individuals will be approached. Relevant NGOs will be contacted to recruit individuals with self-reported physical and mental limitations (e.g. ADHD, anxiety disorder, deafness, blindness) and charity organizations to reach individuals without a permanent residence. Moreover, an advertisement will be published in

social media and in local newspapers from different parts of Sweden to reach lay persons from both rural and urban areas. In addition, the snowball method [44] will be used as a complement, meaning that participants in a study are asked to identify other potential participants. This sampling technique is purposeful when recruiting from groups considered vulnerable. Different laypersons will be included in the steering group and the three phases of the study.

**Phase 2- Description of the outcome evaluation.** *Participants and study setting*. The outcome evaluation will include all individuals entering the iDA.

*Procedure and outcomes*. After the beta-testing the iDA will be public and accessible online from a laptop, smartphone or padlet, for all who are invited to and/or interested in the cancer screening programmes. In order to evaluate and further develop the iDA, topics such as views on the iDA, including suggestions of improvements will be collected together with digital health literacy, knowledge, values and preferences, decisional conflict, concerns and risk awareness will be studied by means of self-reported questionnaires. In addition, demographic data regarding sex, age, educational level and employment will be included as well as questions about lifestyle and physical activity. Usability of the DA including the chat function, helpline and behaviour flow, i.e., how individuals navigate in the iDA, will also be scrutinized.

*Data collection*. Data will be collected through the self-reported questionnaires included in the iDA (S2 Fig. Self-reported questionnaires phase 2). For phase 2, informed consent will be obtained within the digital DA by ticking a box (written).

*Data management*. Data will only be extracted and analysed for those who consent and all names will be removed and participants given a unique code. All data will be stored in a secure facility at the Division of Nursing, Karolinska Institutet in accordance with university regulations.

*Data analyses*. Data from the iDA, including usability and behaviour flow will be identified and analysed on group level, e.g. number of times different parts of the DA were visited and navigation patterns. Data from self-reported questionnaires such as, demographic data, knowledge, values and preferences etc. will be analysed by applying relevant statistical procedures to the data generated, such as between-and-within-group comparisons and regression analyses with relevant demographic aspects as independent variables.

*Participant timeline*. Phase 2. During 2026 and onwards: implementation of the DA including an on-going evaluation.

*Recruitment*. All individuals entering the iDA will be approached and asked for permission to save their demographic and questionnaire data. Everyone who consents will be included in the study.

## Ethics

The risk of violation of integrity is considered small as data will be analysed and presented on group level. All individuals included in phase 1 will sign a consent form prior to inclusion. For phase 2, informed consent will be obtained within the DA. This project has been reviewed by The Swedish Ethical Review Authority, Dnr 2022-00786-01 and the authority had no ethical objections.

## Discussion

The aim of the present project is to develop and implement a digital iDA for individuals invited to cancer screening in Sweden. The iDA is intended to contribute so that individuals invited to screening base their decision on knowledge and with a clear picture of their values and preferences, rather than ignorance, misconceptions, or fear. Furthermore, the iDA is

expected to increase knowledge and raise general awareness about cancer and cancer screening.

To the best of our knowledge the present study is the first attempt aimed at developing such an iDA for use in a Swedish screening context, although similar projects, such as applying a family-based multimedia intervention in relation to CRC screening, have been undertaken elsewhere [45]. Another, recently published study presented 23 conclusions formulated and voted on by participants taking part in a public deliberation regarding the design and evaluation of a DA focusing on CRC screening [37]. Conclusions included the Usage of a DA, the Content (encouraging screening, supporting informed decision making), (complexity and tailoring), (quantitative information), and Trustworthiness. These 23 conclusions can serve as a reminder for us during the developmental process, together with the following statements made by participants: keep DAs short and simple, support participants' engagement and understanding and tailor DAs to each individual user.

Applying the IPDAS quality criteria as a basis [29, 31] and including the model development process for DAs [30] are considered a strength, contributing to the proper development of the DA. The criteria are well-known, thoroughly developed [30, 31] and previously used in similar settings and for similar purposes [35]. Further advantages are the inclusion of a gender and diversity perspective and the mapping onto the SDGs Good health and well-being, Gender equality and Reduced inequalities, all included in Agenda 2030 [22]. Approaching individuals who normally do not participate in cancer screening, or research in general, for that matter, will hopefully contribute to a successful design and development process with the intention of making the iDA inclusive and accessible for as many as possible. This, regardless of any physical or mental limitation or ethnic background. The iDA is considered to be an informative and interactive tool and since it will be digital, the behaviour flow can be studied, meaning that instant adaptions and modifications of content and structure can be performed continuously. The goal is to have more individuals making autonomous and informed decisions, regarding cancer screening participation, and in line with their values and preferences. The Swedish national screening programmes are directed towards three common diseases, preventable and treatable through preventive efforts, and where a decision aid is well-suited. It is desired that individuals make an informed decision on knowledge rather than ignorance, misconceptions, or fear. Therefore, the present project is performed in close collaboration with stakeholders, experts and end-users. The latter including individuals with diverse ethnic backgrounds and limitations, a composition rare in research, nevertheless of importance from an equity perspective.

Nevertheless, the project still faces a couple of challenges. Firstly, the recruitment of lay persons for the developmental process might be difficult. It is common knowledge these days that recruiting people for research purposes is challenging. We will try to resolve this issue by advertising in social media and local newspapers in different parts of Sweden. However, we believe that people will find it rewarding to take part in this project by contributing with their opinions and thoughts. A small incentive will be offered to each individual (two cinema tickets).

Secondly, internet access in Sweden is generally good and the vast majority, regardless of age, use and search for information on the internet on a regular basis. However, it will be a challenge to reach out to those who do not use the internet and/or without a permanent residence. In order to address this challenge we will approach NGOs and charity organizations in addition to applying the snowball method [44]. Furthermore, the iDA will initially be paper based during the design phase, thus the challenge of internet access will only emerge later in the process by which time it will hopefully be solved by the provision of internet access by the specific NGO or charity organization.

## Supporting information

**S1 Checklist. SPIRIT checklist.**
(DOC)

**S1 Fig. An overview of the process phase 1.**
(DOCX)

**S2 Fig. Self-reported questionnaires phase 2.**
(DOCX)

**S1 File. Project plan as approved by the The Swedish Ethical Review Authority.**
(DOCX)

## Author Contributions

**Conceptualization:** Kaisa Fritzell, Berith Hedberg, Anke Woudstra, Anna Forsberg, Marika Sventelius, Anders Kottorp, Anna Jervaeus.

**Investigation:** Kaisa Fritzell, Anna Jervaeus.

**Methodology:** Kaisa Fritzell, Berith Hedberg, Anke Woudstra, Marika Sventelius, Anna Jervaeus.

**Project administration:** Kaisa Fritzell, Anna Jervaeus.

**Validation:** Anna Forsberg, Marika Sventelius.

**Writing – original draft:** Kaisa Fritzell, Berith Hedberg, Anke Woudstra, Anna Forsberg, Marika Sventelius, Anders Kottorp, Anna Jervaeus.

**Writing – review & editing:** Kaisa Fritzell, Berith Hedberg, Anke Woudstra, Anna Forsberg, Marika Sventelius, Anders Kottorp, Anna Jervaeus.

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
