## [Decision Letter · Decision Letter 0]

20 Jul 2023

PONE-D-23-16494Making the BEST decision-the BESTa project

Development and implementation of a digital Decision Aid in Swedish cancer screening programmes- a Study ProtocolPLOS ONE

Dear Dr. Jervaeus,

Thank you for submitting your manuscript to PLOS ONE. After careful consideration, we feel that it has merit but does not fully meet PLOS ONE’s publication criteria as it currently stands. Therefore, we invite you to submit a revised version of the manuscript that addresses the points raised during the review process.

ACADEMIC EDITOR: 

Thank you for giving us the opportunity to consider your manuscript. It has now been reviewed and the reviewers' comments are listed below. In particular, one reviewer has expressed considerable criticism of the nature of the article because, even though you declare it to be a study protocol, much basic information is missing and the manuscript as it stands is not sufficient to fully understand what you are doing and how to replicate your intervention elsewhere. Please carefully consider the comments made.

We will consider publishing your manuscript only if you can incorporate the suggestions in a revised version or satisfactorily explain why the comments are not applicable.

We look forward to receiving your revised manuscript.

Kind regards,

Laura Brunelli, MD

Academic Editor

PLOS ONE

Journal Requirements:

5. We note that Supporting information Appendix 1 and 2 in your submission contain copyrighted images. All PLOS content is published under the Creative Commons Attribution License (CC BY 4.0), which means that the manuscript, images, and Supporting Information files will be freely available online, and any third party is permitted to access, download, copy, distribute, and use these materials in any way, even commercially, with proper attribution. For more information, see our copyright guidelines: http://journals.plos.org/plosone/s/licenses-and-copyright.

a. You may seek permission from the original copyright holder of Supporting information Appendix 1 and 2 to publish the content specifically under the CC BY 4.0 license. 

Additional Editor Comments:

Thank you for giving us the opportunity to consider your manuscript. It has now been reviewed and the reviewers' comments are listed below. In particular, one reviewer has expressed considerable criticism of the nature of the article because, even though you declare it to be a study protocol, much basic information is missing and the manuscript as it stands is not sufficient to fully understand what you are doing and how to replicate your intervention elsewhere. Please carefully consider the comments made.

We will consider publishing your manuscript only if you can incorporate the suggestions in a revised version or satisfactorily explain why the comments are not applicable.

Reviewers' comments:

Reviewer's Responses to Questions

**Comments to the Author**

1. Does the manuscript provide a valid rationale for the proposed study, with clearly identified and justified research questions?

Reviewer #1: Yes

Reviewer #2: Partly

2. Is the protocol technically sound and planned in a manner that will lead to a meaningful outcome and allow testing the stated hypotheses?

Reviewer #1: Yes

Reviewer #2: Partly

3. Is the methodology feasible and described in sufficient detail to allow the work to be replicable?

Reviewer #1: Yes

Reviewer #2: No

4. Have the authors described where all data underlying the findings will be made available when the study is complete?

Reviewer #1: Yes

Reviewer #2: No

5. Is the manuscript presented in an intelligible fashion and written in standard English?

Reviewer #1: Yes

Reviewer #2: Yes

6. Review Comments to the Author

You may also provide optional suggestions and comments to authors that they might find helpful in planning their study.

Reviewer #1: Thank you for the opportunity to review this protocol paper on the development of a digital decision aid for cancer screening in the Swedish context, a timely study given the rising rates of cancer across the globe. There are, however, some gaps that need to be addressed to strengthen the paper:

Background:

• The authors describe the low uptake of cancer screening in general. However, more contextual data is needed about Sweden. What is the current uptake of cancer screening, across cancer programs and demographics?

• It is unclear how the current screening program works in Sweden. What is meant by ‘region’? Is it a local health department that sends the invitations for cancer screening?

Methods

• The authors focus on the issue of equity and how this digital tool will aim to reduce ongoing inequities in cancer screening. However, more information is needed on how this will be achieved.

o For example, the study will aim to recruit individuals with physical and mental limitations. What are examples of these limitations? Furthermore, how will the study procedures, staff and ultimately, the digital tool accommodate the needs of these participants?

o Furthermore, it would be helpful if the authors could expand on the diversity of participants they aim to recruit, whether they will recruit non-English speaking participants and what translation services will be provided if any.

• The authors describe the clinical experts that will be recruited throughout the study. Can they speak on whether any experts in decision-making and digital tools will be included?

• A more in depth description of the qualitative methods that will be used is needed. In particular, it would be helpful if the authors expand on how the data will be analysed and synthesized between the different stages of the study and participants. Additional details on the descriptive and manifest content analysis that is mentioned is also needed.

• 33 participants will be included for each screening program interview study. How was this number selected?

• The flow diagram in the appendix would benefit from additional details such as the number of participants included at each stage.

Discussion

• The authors mention that patients and the lay public are involved in the study. However it is a little unclear about whether this is related to the direct study participation or actual design of the protocol.

• Some details around how long will the decision aid take users to complete would be very helpful.

Other

• There is a small typo in Appendix 1- should be ‘alpha’ not ‘alfa’

Reviewer #2: Although I have concluded that a major revision is required, I am still rather mystified by the tittle including the word ‘Protocol’. Perhaps what is needed is a rethink as to the purpose of this paper?

Simon Day defined a Protocol as “written document describing all the important details of how as study will be conducted. It will generally include details of the product’s being used, rationale for the study, what procedures will be carried out on subjects in the study, how many subjects will be studied, the design of the study and how the data will be analysed”

Clearly although this paper is not describing a clinical trial, it is allegedly describing a study which will require substantial and detailed blocks of activities requiring how and when the various components will be implemented. So, a protocol is more like an instruction manual which the current paper clearly is not.

From a statistical perspective there is, for example, no indication of how many subjects will be interviewed at the various stages, nor of how the overall assessment of the value of the programme will be made.

Reference: Day S (2007) Dictionary for Clinical Trials. Wiley, Chichester

7. PLOS authors have the option to publish the peer review history of their article (what does this mean?). If published, this will include your full peer review and any attached files.

Reviewer #1: No

Reviewer #2: No

---

## [Author Response · Author response to Decision Letter 0]

25 Sep 2023

Please see uploaded file regarding reviewer and editor comments.

---

## [Decision Letter · Decision Letter 1]

30 Oct 2023

Making the BEST decision-the BESTa project

Development, implementation and evaluation of a digital Decision Aid in Swedish cancer screening programmes- a description of a research project

PONE-D-23-16494R1

Dear Dr. Jervaeus,

We’re pleased to inform you that your manuscript has been judged scientifically suitable for publication and will be formally accepted for publication once it meets all outstanding technical requirements.

Kind regards,

Laura Brunelli, MD, PhD

Academic Editor

PLOS ONE

Additional Editor Comments (optional):

Reviewers' comments:

Reviewer's Responses to Questions

**Comments to the Author**

1. Does the manuscript provide a valid rationale for the proposed study, with clearly identified and justified research questions?

Reviewer #2: Yes

2. Is the protocol technically sound and planned in a manner that will lead to a meaningful outcome and allow testing the stated hypotheses?

Reviewer #2: Yes

3. Is the methodology feasible and described in sufficient detail to allow the work to be replicable?

Reviewer #2: Yes

4. Have the authors described where all data underlying the findings will be made available when the study is complete?

Reviewer #2: Yes

5. Is the manuscript presented in an intelligible fashion and written in standard English?

Reviewer #2: Yes

6. Review Comments to the Author

You may also provide optional suggestions and comments to authors that they might find helpful in planning their study.

Reviewer #2: The authors have addressed my earlier concerns in their revision.

7. PLOS authors have the option to publish the peer review history of their article (what does this mean?). If published, this will include your full peer review and any attached files.

Reviewer #2: No

---

## [Editor Report · Acceptance letter]

1 Dec 2023

PONE-D-23-16494R1 

Making the BEST decision-the BESTa project
Development, implementation and evaluation of a digital Decision Aid in Swedish cancer screening programmes- a description of a research project 

Dear Dr. Jervaeus:

I'm pleased to inform you that your manuscript has been deemed suitable for publication in PLOS ONE. Congratulations! Your manuscript is now with our production department. 

Kind regards, 

on behalf of

Dr. Laura Brunelli 

Academic Editor

PLOS ONE